# Childhood Disability and Nutrition: Findings from a Population-Based Case Control Study in Rural Bangladesh

**DOI:** 10.3390/nu11112728

**Published:** 2019-11-11

**Authors:** Israt Jahan, Tasneem Karim, Mahmudul Hassan Al Imam, Manik Chandra Das, Khaled Mohammad Ali, Mohammad Muhit, Gulam Khandaker

**Affiliations:** 1CSF Global, Dhaka 1213, Bangladesh; arda.jahan89@gmail.com (I.J.); tasneem.karim.tk@gmail.com (T.K.); physiomahmud@yahoo.com (M.H.A.I.); manikchandradas10@gmail.com (M.C.D.); khaled.ali.kma@gmail.com (K.M.A.); mmuhit@hotmail.com (M.M.); 2Asian Institute of Disability and Development (AIDD), University of South Asia, Dhaka 1213, Bangladesh; 3School of Health, Medical and Applied Sciences, Central Queensland University, Rockhampton, Queensland 4701, Australia; 4Discipline of Child and Adolescent Health, Faculty of Medicine and Health, The University of Sydney, Sydney 2145, Australia; 5Cerebral Palsy Alliance Research Institute, The University of Sydney, Sydney 2050, Australia; 6Central Queensland Public Health Unit, Central Queensland Hospital and Health Service, Rockhampton, Queensland 4700, Australia

**Keywords:** childhood disability, malnutrition, epidemiology, key informant method, Bangladesh

## Abstract

Background: Evidence regarding the complex relationship between childhood disability and malnutrition is limited in low and middle income countries. We aimed to measure the association between childhood disability and malnutrition in rural Bangladesh. Method: We conducted a population-based case control study among children aged <18 years in a rural sub-district (i.e., Shahjadpur) in Bangladesh. Children with permanent disability (i.e., Cases) and their age/sex-matched peers (i.e., Controls) were identified from the local community utilizing the key informant method. Socioeconomic, anthropometric, and educational information was collected using a pre-tested questionnaire. Only Cases underwent detailed medical assessment for clinical and rehabilitation information. Descriptive and bivariate analyses were performed. Results: Between October 2017 and February 2018, 1274 Cases and 1303 Controls were assessed. Cases had 6.6 times and 11.8 times higher odds of being severely underweight and severely stunted respectively than Controls. Although epileptic children had the highest overall prevalence of malnutrition, the age/sex-adjusted odds of malnutrition were significantly higher among children with physical impairments. Underweight and/or stunting among children with disability was significantly associated with parental educational qualification, socioeconomic status and mainstream school attendance. Conclusion: The significantly high proportion of severe malnutrition among children with disability calls for urgent action and implementation of inclusive nutrition intervention programs in rural Bangladesh.

## 1. Introduction

Childhood disability and malnutrition are two major public health concerns that play a crucial role in health-related outcome, quality of life and survival. Globally, it is estimated that 15.6% of the world’s population have at least some form of disability and 80% of them live in low and middle income countries (LMICs) [1]. However, the true burden of childhood disability is yet unknown and existing evidence suggests that there are approximately 93 to 150 million children aged 0–18 years living with disability worldwide [2].

Although the global burden of malnutrition has decreased in the past decade, millions of children aged less than five years still strive to break the cycle of malnutrition. In 2018, it was estimated that 150.8 million children aged less than five years are stunted (22.5%), and 50.5 million children are wasted (7.5%) [3]. With this high burden, malnutrition still remains a leading cause of childhood mortality and the scenario becomes complex when malnutrition is coupled with disability, especially in LMICs. The recent estimates suggest that malnutrition accounts for 45% of all childhood death—3.1 million deaths each year among children globally [4].

Malnutrition plays a crucial role as both the cause and consequence of childhood disability [2,5]. In a recent systematic review, the pooled analyses showed that children with disability have 3.0 times higher odds of being underweight compared to children without disability [6]. In another study conducted in Kenya, it was found that children with disability had 2.2 times higher odds of being underweight and 1.7 times higher odds of being stunted when compared to their age/sex-matched peers who did not have any forms of permanent disability [7].

Poor socioeconomic conditions and multiple impairments could result in poorer health and development outcomes, subsequently leading to a perpetuating cycle of suboptimal nutrition, disability and worsening health status. Although several studies and global organizations have developed models illustrating the factors interlinked with the occurrence and progression of disability among children, the direct pathways vary depending on the type of disability. In LMICs such as Bangladesh, inadequate service provision, a lack of need-based rehabilitation and, most importantly, a lack of evidence-based intervention programs makes it difficult for children with disability to maintain a standard life. In one study, in Bangladesh, it was found that only 24% of children aged 2–4 years with disability had received vitamin A supplementation in the previous 6 months [8].

However, such evidence is scarce and there are still unexplored opportunities to study the complex relationship between childhood disability and malnutrition in Bangladesh and other LMICs. In this study, we aimed to measure the association between childhood disability and malnutrition and identify the role of socioeconomic context and the types and severity of impairments as contributing factors of malnutrition among children with disability in rural Bangladesh.

## 2. Materials and Methods

We conducted a population-based case control study among children aged <18 years living in one of the northern subdistricts (i.e., Shahjadpur) of Bangladesh. The study area covers 324 km^2^, including 296 villages, 123,576 households, and 561,076 people (of whom, ~ 232,037 are children aged <18 years) [9].

### 2.1. Study Participants

We defined (i) Cases as children with permanent disability, according to the definitions of International Classification of Functioning, Disability and Health (ICF) (i.e., physical impairment, visual impairment, hearing impairment, speech impairment, and epilepsy) [10], aged <18 years and living in the study area; (ii) Controls as peers of children with disability (i.e., Cases) who do not have functional impairment, aged <18 years and living in the same community of the Cases.

### 2.2. Identification of Children with Disability

We formed the first population-based cohort of children with disability in rural Bangladesh known as Shahjadpur Children’s Cohort (SCC) through this study. Children with disability were identified and recruited in SCC using the key informant method (KIM) [11,12]. The KIM is a novel method where local trained volunteers (i.e., key informants (KIs)) identify children with possible disability and assist their (i.e., children with disability) families to bring identified children to medical assessment camps for confirmed diagnosis and detailed clinical assessment [11,12].

The study team has previously established two cohorts utilizing the KIM in the study area; (i) Severe Childhood Disability Cohort: an active community-based survey established in 2012 [13], and (ii) Bangladesh Cerebral Palsy Register (BCPR): an ongoing surveillance of children with cerebral palsy (CP) established in 2015 [14,15].

As part of this study, two community mobilizers (CMs—paid project staff) identified 130 KIs, focusing on geographical coverage of the study area. The KIs were then provided with one-day training on the identification of children with different types of impairments (suspected Cases) and general counselling to mitigate negative attitude toward disability in the community. With help of the KIs, all eligible children with disability from the two cohorts mentioned above were retraced. At the same time, if new children with suspected disability were identified in the communities, they were also brought to the medical assessment camps. Children who had confirmed diagnosis of any type of disability strictly following the ICF definitions [10] were registered in the SCC.

### 2.3. Selection of Controls

Individually age (±5 years) and sex-matched Controls were selected randomly from the study area. Each KI identified two Controls who met the following criteria; (i) does not have any form of disability, (ii) age/sex matched, (iii) living in the same community (approximately within 10–20 households) of the child identified with suspected disability. The KIs were unaware of any individual study characteristics (except for age and sex) during the selection of controls.

### 2.4. Data Collection

Each of the children with disability, i.e., Cases (retraced and newly identified from community) underwent detailed clinical assessment for confirmed diagnosis of impairments by a medical assessment team including a pediatrician, ophthalmologist, and physiotherapist, who recorded detailed information on clinical characteristics (e.g., types of impairments and severity) and rehabilitation status. Detailed information on socio-demographic characteristics, socioeconomic status, nutritional status, and educational status was also collected from both Cases and Controls. All information was collected using a pre-tested standard comprehensive data collection tool.

### 2.5. Anthropometric Measurement

The weight and height of each study participant were measured following standard guidelines of the World Health Organization (WHO) [16].

#### 2.5.1. Weight Measurement

Weight was measured in kilograms using a digital weighing scale with a precision of 1 gram. Tared weight was measured for young children or children with physical deformities who could not stand independently. Three repeated measures were taken, and the average was documented. The weighing scale was set to zero each time before taking a new measurement.

#### 2.5.2. Height Measurement

Length/height was measured in centimeters (cm) using a measuring tape with a precision of 1 cm placed on a length board or height board. Recumbent length was measured for children aged less than two years using a length board and height was measured for children aged two years and above using a height board. For children with physical deformities who could not stand independently, segmental measurement (i.e., knee height) was measured and height was estimated using the formula Height = (2.69 × knee height) + 24.2 cm [17].

## 3. Statistical Analysis

We performed descriptive analyses to estimate the prevalence of malnutrition with a 95% CI among children with disability. We also described the socioeconomic characteristics of study participants (i.e., children with disability and children without disability) and compared the profile of children with disability to national data. Bivariate analyses were carried out to identify factors related to malnutrition among children with disability. Adjusted analyses were performed to identify potential predictors of underweight and stunting. Age- and sex-adjusted odds ratios were calculated for different forms of malnutrition among children with disability and children without disability. All analyses were conducted using SPSS Statistics software version 23 (IBM, Armonk, NY, USA).

To measure the nutritional status of study participants, anthropometric measurements were compared to WHO standards for the child population and malnutrition was defined using three different indices, namely (i) weight for age (WAZ) as an indicator of general malnutrition; (ii) height for age (HAZ) as an indicator of chronic malnutrition; (iii) weight for height (WHZ) as an indicator of acute malnutrition. All *z* scores were calculated using WHO Anthro and WHO AnthroPlus software. Malnutrition was defined following WHO standards based on the *z* scores as (i) general malnutrition (WAZ <−2SD and >2SD), (ii) chronic malnutrition (HAZ <−2SD and >2SD) and (iii) acute malnutrition (WHZ <−2SD and >2SD) [16]. A child was considered moderately undernourished if WAZ and/or HAZ and/or WHZ <−2SD and ≥−3SD and severely undernourished if WAZ and/or HAZ and/or WHZ <−3SD [16].

### Ethical Consideration

Ethical approval was taken from the Bangladesh Medical Research Council (BMRC) (BMRC/NREC/2016-2019/468) and Asian Institute of Disability and Development (AIDD) (HREC ref no: southasiairb-2017-8-01). Informed written consent was given by the primary caregivers of each of the study participants (following an explanation of the study objectives, the voluntary nature of participation, and the confidential handling of information) before collecting information.

## 4. Results

Between October 2017 and February 2018, 1274 children with disability and 1303 children without disability were assessed. According to the 2011 census, there are an estimated 232,037 children aged less than 18 years living in the study area (i.e., Shahjadpur subdistrict), which gives a minimum disability prevalence of 0.55% (95% CI 0.52–0.58%) in Shahjadpur subdistrict. The mean (SD) age at assessment of children with disability and children without disability was 9.7 (4.7) years and 9.0 (5.2) years, respectively. A total of 56.4% (*n* = 718) Cases and 46.7% (*n* = 608) Controls were male. The major types of impairment in the SCC cohort were physical impairment (71.8%, 95% CI 69.3–74.2%), followed by speech impairment (66.9%, 95% CI 64.3–69.5%) and hearing impairment (30.6%, 95% CI 28.1–33.2%). Among children with physical impairment, 87.0% (*n* = 796) had CP. The prevalence of physical, visual, hearing and speech impairment was significantly higher among our study cohort compared to national data (*p* < 0.001), (Table 1).

Significant differences in access to sanitary toilet facilities, the educational level of parents, the occupation of parents, socioeconomic status and mainstream school attendance were observed between Cases and Controls. When adjusted for age and sex, children with disability had higher odds of using a hanging toilet (aOR (95% CI): 4.1 (1.6–10.4)); father being unemployed (aOR (95% CI): 4.4 (1.4–14.0)), poor socio-economic status (aOR (95% CI): 1.6 (1.3–2.0)), and not being enrolled in mainstreaming school (aOR (95% CI): 42.2 (29.0–61.3) (Table 2).

### 4.1. Nutritional Status of Children with Disability

Overall, the nutritional status of children with disability was significantly poor compared to the Controls. Children with disability had 6.6 times higher odds of being severely underweight and 11.8 times higher odds of being severely stunted compared to Controls of the same age and sex. A similar pattern was observed for the presence of acute malnutrition among children aged less than five years. When adjusted for age-and sex, the odds of severe acute malnutrition (SAM) were 4.0 times higher among Cases than Controls living in the study area (Table 3).

Figure 1 and Figure 2 illustrate the burden of malnutrition among children of both groups according to their completed years of age and disability status. Overall, the proportion of both underweight and stunting gradually increased until the age of 4 years among children with disability. A similar pattern was observed for children without disability. However, the burden of malnutrition was significantly high in children with disability compared to others in the cohort. Moreover, both acute and chronic malnutrition was found significantly overrepresented among young children with disability aged less than five years compared to children without disability (*p* < 0.001). However, for older children aged over five years, the deviations in nutritional status were inconsistent disregarding the disability status of the study participants.

The estimated prevalence of underweight and stunting was highest among children with epilepsy (75.9%, *n* = 82/108 and 79.4%, *n* = 139/175, respectively) followed by children with visual impairment (70.1%, *n* = 54/77 and 73.4%, *n* = 113/154, respectively) and children with physical impairment (69.8%, *n* = 363/520 and 73.7%, *n* = 662/898, respectively). The median (IQR) WAZ and HAZ was poorer among children who had three or more impairments compared to those who had one impairment (−2.7 (−4.0, −1.4) vs. −2.1 (−2.6, −1.0); *p* = 0.080 and −3.0 (−5.2, −1.5) vs. −2.7 (−6.5, −2.2); *p* < 0.001). When adjusted for age and sex, the odds of underweight was significantly higher among children who had physical impairment and/or children who had clinically diagnosed epilepsy. The estimated prevalence of underweight and stunting was 71.2% (95% CI: 67.0–75.0%) and 75.0% (95% CI: 72.0–78.0%), respectively, among children with CP. Furthermore, both underweight and stunting were found high among children with CP who had tri/quadriplegia and/or Gross Motor Function Classification System (GMFCS) level III–V and/or hearing impairment and/or speech impairment. (Table 4 and Table 5).

### 4.2. Relationship between Socioeconomic Characteristics and Malnutrition (i.e., Underweight and/or Stunting)

Table 6 and Table 7 summarize the relationship between socio-demographic characteristics and malnutrition among children with disability and children without disability. When adjusted for age and sex, the odds of being underweight and/or stunting was significantly associated with parental educational level, socioeconomic status and attendance to mainstream school among both children with disability and children without disability. However, the degree of association was higher among children with disability compared to children without disability. 

## 5. Discussion

The study findings have great significance in understanding the association between malnutrition and disability from a population-based case control study. We presented the age/sex-adjusted analysis to exclude the confounding effect of these demographic features on the nutritional status of children. The findings clearly illustrate that growth faltering is common among children with disability and severe malnutrition is frequently observed among them compared to children without disability of same age and sex, living in rural Bangladesh.

The interaction between disability and malnutrition has been reported repeatedly in different studies [5,6,7]. In their recent systematic review, Hume-Nixon et al. reported a 53% positive association between childhood disability and malnutrition in LMICs [6]. The review included 17 studies from LMCIs and pooled analysis showed three times higher odds of underweight and two times higher odds of stunting and wasting among children with disabilities compared to controls [6]. However, according to the authors, due to a limited number of studies on this topic from LMICs, the effect of the underlying factors on association between malnutrition and childhood disability could not be measured [6]. Furthermore, the lack of representation of all essential types of disability (e.g., visual impairment) among the available studies also interfered in establishing the potential role of ‘types of disability’ on malnutrition prevalence among children in LMICs [6]. In our study, we estimated the prevalence of malnutrition among children separately for major types of disability from a population-based matched case control study in rural Bangladesh. The same systematic review by Hume-Nixon et al. also reported an underrepresentation of female participants and poor quality of evidence among the limited available studies which posed a concern in exploring the gender role in malnutrition among children with disability in LMICs [6]. In our study, we have included 556 female participants with disability, making this one of the largest cohorts of females with disability in LMICs. In addition, we have included equally matched female controls in our study. This allowed us to estimate the age- and sex-adjusted prevalence of malnutrition among children with disability living in rural Bangladesh. Furthermore, we estimated the effect of the potential contributing factors of malnutrition from an age/sex-matched population-based case control study in rural Bangladesh. The study findings are therefore generalizable to a broader population and are crucial for program planners as well as policy makers to implement disability-inclusive intervention programs in Bangladesh and other LMICs.

Malnutrition can act as a cause and also as a consequence of disability and thus have intergenerational implication if not addressed properly [5]. Evidence from different countries have identified several underlying risk factors for malnutrition among children with disability [5]. However, in this population-based case control study, we focused on the empirical role of the socioeconomic context in growth retardation among children with disability in LMICs.

We found a significant association of poor socioeconomic status with malnutrition among children with disability. The role of poor socioeconomic status is both direct and indirect. Children with disability often face difficulties in maintaining adequate nutritional intake, mostly associated with the severity of impairments, and require specialized assistance and arrangements for better nutrition. Our study findings suggest that the majority of families of children with disability had a monthly family expenditure higher than their respective incomes, which indicates that the families were more likely to be in debt. We also observed that the parents of children with disability were more likely to be involved in blue-collar jobs and/or were unemployed compared to the age/sex-matched Controls. This suggests that these families, due to their financial hardship, might have lacked the means for special efforts or access facilities to correct the nutritional inadequacy of their children with disability, which in the long run might have led to nutritional imbalance and reduced growth.

On the other hand, children living in poor socioeconomic contexts are most likely to have insufficient access to improved water, sanitation and hygiene (WASH), educational attainment and health care services [18]. In our study, children with disability had significantly higher levels of poor hygiene practices and poor maternal educational level compared to Controls. We also found a significant positive association between malnutrition and poor sanitation practices among children with disability. Poor WASH practices increase the susceptibility of infection (e.g., bacterial and parasitic) and malnutrition [19]. In a recent systematic review, it was reported that 20% of protein-energy malnutrition among children is attributed to poor WASH practices, and it was also estimated that 62.2% of all diarrheal deaths that occurred globally in 2016 were due to inadequate WASH practice [19].

As discussed earlier, we estimated the magnitude of malnutrition among children with disability according to their type of impairments. The age/sex-adjusted odds ratio indicates that children with physical impairment living in the study site were more vulnerable to malnutrition compared to children with other forms of disability (*p* < 0.05). One study conducted in Iran reported that children with physical impairment had poor dietary intake and the majority did not meet half of the recommended dietary allowances for different micronutrients [20]. In our study, among children with physical impairment, the majority had CP, a neurodevelopmental disorder that affects the motor function of children [21]. The high burden of malnutrition among children with CP has been reported in several studies previously [22,23]. Evidence also suggests that children with CP are vulnerable to feeding difficulties, gastroesophageal reflux and metabolic alterations, which interfere with adequate nutritional management and lead to poor nutritional outcome [21,22,23,24]. Our study findings reveal that children with CP who had severe forms of motor impairments had significantly higher odds of becoming malnourished. Similar findings were reported in other studies conducted among children with CP living in LMICs [22,23,24,25].

We also found a significant association between malnutrition and clinically confirmed diagnosis of epilepsy in our SCC cohort. A review exploring the relationship between epilepsy and malnutrition reported that early childhood malnutrition, such as protein energy malnutrition or the deficiency of certain micronutrients, can impair regular central nervous system (CNS) activities and is related to the development of epilepsy among children [26]. Alternatively, growth faltering and malnutrition has also been reported in several studies conducted among children with epilepsy, mostly attributed to poor dietary intake, the effect of several antiepileptic drugs and changes in metabolism [26].

Our study findings show that children with disability were not only vulnerable to malnutrition, but also lacked mainstream school attendance. Moreover, children who lacked mainstream school attendance had significantly higher odds of becoming malnourished compared to those who were enrolled in schools. It is possible that children with disability who were attending the schools had less severe forms of impairment and did not require special assistance for attendance. Furthermore, they might have also participated in different school-based nutrition interventions/feeding programs.

Although the findings have important implications on understanding the complex relationship between childhood disability and malnutrition, the study had several limitations. First, our estimated prevalence of disability and malnutrition among children with disability could be an underestimation as we utilized the KIM in identifying children with disability from the community. However, the KIM is a cost-effective method compared to a door-to-door survey (77.6% case ascertainment for all cause disability at 25% cost of door-to-door survey) [12,27], has 98% sensitivity for the identification of children with disability from the community, and it is therefore highly effective in the resource-poor settings of LMICs [12]. Second, the indicators we used (e.g., weight and height) have limitations in reflecting the nutritional status of children with disability precisely and use of other forms of measurements (e.g., skin-fold thickness) could help in triangulation of data and present detailed estimations of malnutrition among children with disability. However, considering the high number of coverage and basic resources available in rural settings for children aged <18 years, these were the optimal choices. Third, the *z* scores were calculated using WHO Anthro software (for WHZ) and WHO AnthroPlus software (for WAZ and HAZ). As WHO Anthro calculates WHZ for children aged 0–61 months and WHO AnthroPlus calculates WAZ for children aged 0–121 months only, we could not determine the presence of general malnutrition (WAZ <−2SD) and acute malnutrition (WHZ <−2SD and >2SD) for all children included in the study. Fourth, we could not establish the causal relationship between early childhood disability and malnutrition as we mostly focused on the basic causes and underlying causes of malnutrition but could not collect information regarding the immediate causes (e.g., dietary intake, comorbidities and disease status) as part of the study. Fifth, despite considerable effort, our Cases and Controls were not completely age and sex matched, and there were minimal differences in terms of these characteristics. However, we used age- and sex-adjusted analyses when estimating the prevalence and presenting the associations.

## 6. Conclusions

The study findings reflect population-based data, hence the results are generalizable to a broader group of population living in rural communities in Bangladesh. The significantly high proportion of severe malnutrition among children with disability calls for urgent action to initiate and implement inclusive nutrition intervention programs in rural Bangladesh. The findings should be taken under consideration in the development of strategies and program planning as well as at policy level for better health, quality of life and survival probability of children with disability.

## Figures and Tables

**Figure 1 nutrients-11-02728-f001:**
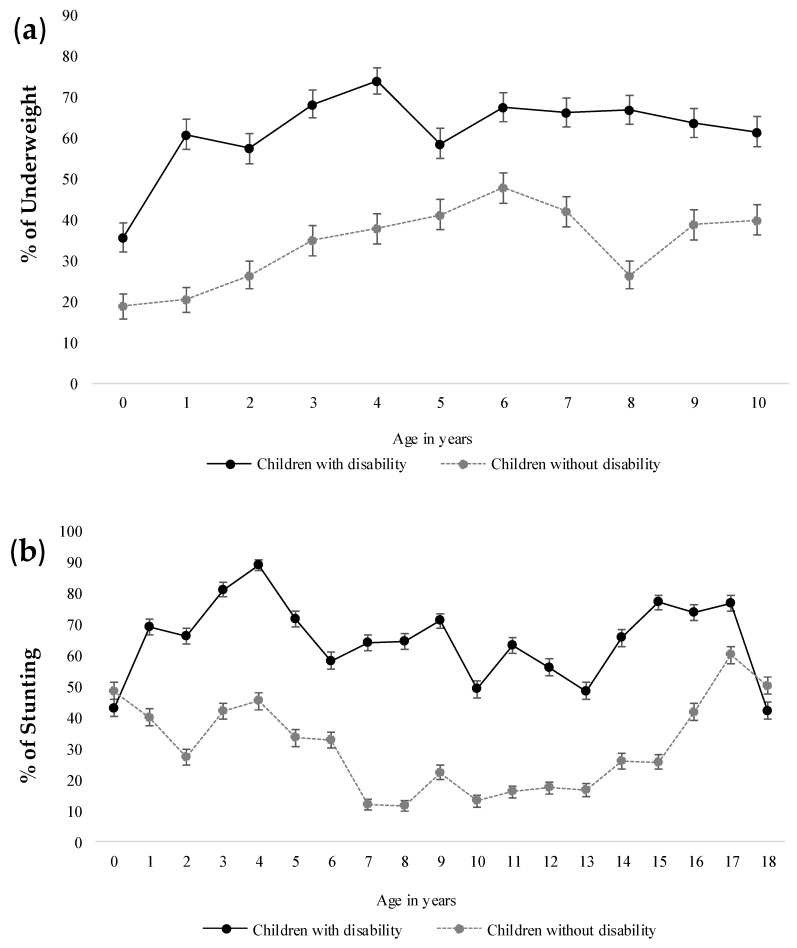
(**a**) Comparison of underweight (%) among children with disability and children without disability according to their age (years). (**b**) Comparison of stunting (%) among children with disability and children without disability according to their age (years).

**Figure 2 nutrients-11-02728-f002:**
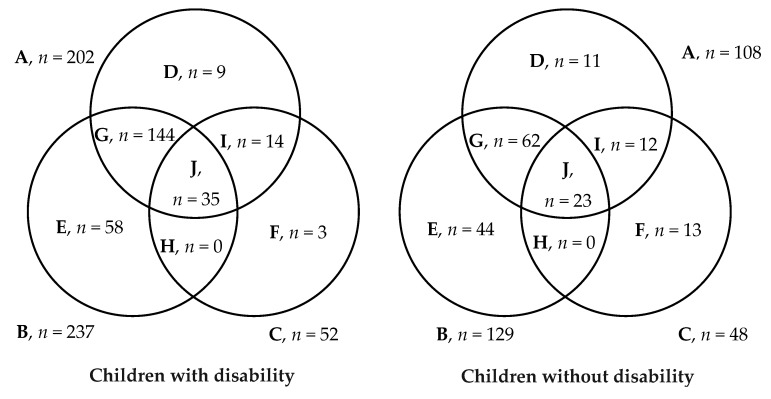
The nutritional status of study participants aged less than 61 months. Here, (**A**) indicates the total number of underweight children (A = D + G + I + J); (**B**) indicates the total number of stunted children (B = E + G + H + J); (**C**) indicates the total number of wasted children (C = F + H + I + J); (**D**) indicates the total number of children who were underweight only; (**E**) indicates the total number of children who were stunted only; (**F**) indicates the total number of children who were wasted only; (**G**) indicates the total number of children who were underweight and stunted but not wasted; (**H**) indicates the total number of children who were stunted and wasted but not underweight; (**I**) indicates the total number of children who were underweight and wasted but not stunted; (**J**) indicates the total number of children who were underweight, stunted and wasted.

**Table 1 nutrients-11-02728-t001:** Characteristics of the study participants.

Characteristics	Children with Disability, *n* (%)	Children without Disability, *n* (%)	General Population (%)
	*n* = 1274	*n* = 1303	
**Age Group (Years)**			
0–4	250 (19.6)	286 (21.9)	26.4 ^1^
5–9	413 (32.4)	389 (29.9)	31.8 ^1^
10–14	408 (32.0)	538 (41.3)	29.1 ^1^
15–18	203 (15.9)	90 (6.9)	12.7 ^1^
**Sex**			
Male	718 (56.4)	608 (46.7)	51.8 ^1^
Female	556 (43.6)	695 (53.3)	48.2 ^1^
**Household Income and Expenditure**			
Median (interquartile range) monthly household income, BDT ~ USD	7000 (6000,10,000) ~ 83 (71,119)	8000 (6000,12,000) ~ 95 (71,142)	13,353 ^2^ ~ 159
Median (interquartile range) monthly household expenditure, BDT ~ USD	7000 (5500,10,000) ~ 83 (65,119)	7500 (6000,10,000) ~ 89 (71,119)	14,156 ^2^ ~ 168
**Types of Disability**			
Physical impairment	915 (71.8)	N/A	36.4 ^3^
Cerebral palsy (CP)	796 (87.0)	N/A	N/A
Birth defect	68 (7.4)	N/A	N/A
Genetic disease-related physical impairment	32 (3.5)	N/A	N/A
Trauma/injury-related physical impairment	12 (1.3)	N/A	N/A
Musculoskeletal dystrophy/atrophy	7 (0.8)	N/A	N/A
Visual	154 (12.1)	N/A	10.8 ^3^
Hearing	390 (30.6)	N/A	6.4 ^3^
Epilepsy	179 (14.1)	N/A	N/A
Speech	853 (66.9)	N/A	21.9^3^

^1^ Population and Housing Census 2011, Bangladesh. ^2^ Household Income and Expenditure Survey (HIES), 2016. ^3^ Disability in Bangladesh: Prevalence and Pattern, 2011.

**Table 2 nutrients-11-02728-t002:** Relationship between disability and socioeconomic characteristics.

Characteristics	Children with Disability, *n* (%)	Children without Disability, *n* (%)	General Population, %	*p* Value ^1^	*p* Value ^2^	Adjusted Odds Ratio ^7^ (95% CI)
	*n* = 1274	*n* = 1303				
**Source of Drinking Water**						
Improved source ^4^	1272 (99.8)	1300 (100.0)	98.2 ^3^	0.364	**<0.001**	N/A
Unimproved source ^4^	2 (0.2)	0 (0.0)	0.8 ^3^	0.364	**<0.001**	N/A
**Water Treatment Method (*n* = 1274)**						
Appropriate ^5^	163 (12.8)	156 (12.0)	9.7 ^3^	0.202	**<0.001**	Ref
Inappropriate	1111 (87.2)	1144 (88.0)	N/A-	0.202	N/A	1.1 (0.8, 1.4)
**Access to Sanitation**						
Flush/pour flush to proper disposal system	488 (38.3)	566 (43.6)	15.8 ^3^	**<0.001**	**<0.001**	Ref
Pit latrine with slab	430 (33.8)	465 (35.8)	53.1 ^3^	0.07	**<0.001**	1.1 (0.9, 1.3)
Open pit latrine	333 (26.1)	258 (19.9)	21.8 ^3^	**<0.001**	**<0.001**	**1.5 (1.2, 1.8)**
Hanging toilet	20 (1.6)	6 (0.5)	3.1 ^3^	**<0.001**	**<0.001**	**4.1 (1.6, 10.4)**
No facility	3 (0.2)	3 (0.2)	3.7 ^3^	0.47	**<0.001**	1.3 (0.2, 6.5)
**Educational Level of Mother (Total Years of Schooling)**						
0	523 (41.1)	499 (38.4)	24.9 ^3^	0.028	**<0.001**	Ref
1–4	185 (14.5)	204 (15.7)	18.0 ^3^	0.131	0.001	0.9 (0.7, 1.1)
5	331 (26.0)	382 (29.4)	11.1 ^3^	**0.004**	**<0.001**	0.9 (0.7, 1.1)
6–9	143 (11.2)	140 (10.8)	31.5 ^3^	0.325	**<0.001**	1.0 (0.8, 1.3)
≥10	92 (7.2)	76 (5.8)	14.4 ^3^	**0.02**	**<0.001**	1.2 (0.9, 1.7)
**Educational Level of Father (Total Years of Schooling)**						
0	608 (47.9)	587 (45.2)	19.2 ^3^	**0.03**	**<0.001**	Ref
1–4	135 (10.6)	155 (11.9)	16.0 ^3^	0.086	**<0.001**	0.8 (0.7, 1.1)
5	231 (18.2)	282 (21.7)	11.3 ^3^	**0.001**	**<0.001**	0.8 (0.7, 1.0)
6–9	131 (10.3)	129 (9.9)	28.1 ^3^	0.323	**<0.001**	1.0 (0.8, 1.3)
≥10	165 (13.0)	145 (11.2)	24.4 ^3^	**0.026**	**<0.001**	1.1 (0.9, 1.5)
**Occupation of Mother**						
Desk job	15 (1.2)	12 (0.9)	6.2 ^3^	**<0.001**	**<0.001**	Ref
Blue-collar job	75 (5.9)	51 (3.9)	25.4 ^3^	**<0.001**	**<0.001**	1.3 (0.5, 3.0)
Business	13 (1.0)	7 (0.5)	14.7 ^3^	**<0.001**	**<0.001**	1.5 (0.4, 5.1)
Agricultural/farming	7 (0.5)	5 (0.4)	5.3 ^3^	**0.002**	**<0.001**	1.2 (0.3, 4.8)
Homemaker	1161 (91.1)	945 (72.5)	N/A	**<0.001**	N/A	1.1 (0.5, 2.4)
Others	3 (0.2)	283 (21.7)	0.4 ^3^	**<0.001**	0.251	0.01 (0.002, 0.034)
**Occupation of Father**						
Desk job	68 (5.3)	82 (6.3)	N/A	0.085	N/A	Ref
Blue-collar job	474 (37.2)	441 (33.8)	N/A	**0.006**	N/A	1.3 (0.9, 1.9)
Business	234 (18.4)	213 (16.3)	N/A	**0.026**	N/A	1.4 (0.9, 2.0)
Agricultural/farming	310 (24.3)	365 (28.0)	N/A	**0.002**	N/A	1.1 (0.7, 1.5)
Unemployed	17 (1.3)	4 (0.3)	N/A	**<0.001**	N/A	**4.4 (1.4, 14.0)**
Others	171 (13.4)	198 (15.2)	N/A	**0.04**	N/A	1.0 (0.7, 1.5)
**Socioeconomic Status (SES)**						
Poor SES	273 (21.4)	225 (17.3)	N/A	**<0.001**	N/A	**1.6 (1.3, 2.0)**
Moderate SES	518 (40.7)	481 (36.9)	N/A	**0.003**	N/A	**1.4 (1.1, 1.6)**
High SES	483 (37.9)	597 (45.8)	N/A	**<0.001**	N/A	Ref
**Monthly Income-Expenditure Balance**						
Negative balance	72 (5.7)	45 (3.5)	N/A	**<0.001**	N/A	**1.7 (1.1, 2.6)**
Equal balance	927 (73.0)	930 (72.1)	N/A	0.236	N/A	1.1 (0.9, 1.4)
Positive balance	270 (21.3)	315 (24.4)	N/A	**0.005**	N/A	Ref
**Mainstream School Attendance**						
Primary	268 (21.2)	460 (35.9)	86.7 ^3,6^	**<0.001**	**<0.001**	Ref
Secondary	67 (5.3)	427 (33.3)		**<0.001**		
Others	28 (2.2)	26 (2.0)		0.32		
No	588 (46.5)	33 (2.6)	N/A	**<0.001**	N/A	**42.2 (29.0, 61.3)**
Not applicable	313 (24.8)	336 (26.2)	N/A	N/A	N/A	N/A

^1^*p* value was calculated using a binomial test to identify significant differences between children with disability and children without disability living in the study area. ^2^
*p* value was calculated using a binomial test to identify significant differences between children with disability and the general population (national data). ^3^ Bangladesh Demographic and Health Survey (BDHS) 2014. ^4^ Source of drinking water has been categorized following the definition of BDHS 2014. Improved source includes piped into dwelling, piped into year/plot, public tap, tubewell or borehole, protected well, rainwater, and bottled water; and unimproved source includes unprotected well, unprotected spring, tanker truck, surface water, and other sources. ^5^ Appropriate water treatment methods include boiling, filtering, bleaching and solar disinfecting. ^6^ Children aged 6–15 years. ^7^ Adjusted for age and sex. The bold font indicates statistically significant findings.

**Table 3 nutrients-11-02728-t003:** Overall nutritional status of study participants.

Indicator	Children with Disability	Children without Disability	*p* Value	Adjusted Odds Ratio ^6^ (95% CI)
**Weight for Age *z* Score (WAZ) (*n* = 1344)**				
*n* ^1^	676	668		
Median (IQR)	−2.7 (−4.0, −1.4)	−1.52 (−2.4, −0.6)	<0.001 ^4^	
Normal	239 (35.4)	434 (65.0)	<0.001 ^5^	Ref
Underweight	144 (21.3)	153 (22.9)		**1.7 (1.3, 2.2)**
Severely underweight	293 (43.3)	81 (12.1)		**6.6 (4.9, 8.9)**
**Height for Age *z* Score (HAZ) (*n* = 2518)**				
*n* ^2^	1249	1269		
Median (IQR)	−2.9 (−5.0, −1.5)	−1.1 (−2.0, −0.0)	<0.001 ^4^	
Normal	429 (34.3)	953 (75.1)	<0.001 ^5^	Ref
Stunted	221 (17.7)	201 (15.8)		**2.3 (1.8, 2.9)**
Severely stunted	599 (48.0)	115 (9.1)		**11.8 (9.3, 14.9)**
**Weight for Height *z* Score (WHZ) (*n* = 510)**				
*n* ^3^	241	259		
Median (IQR)	−0.74 (−1.7, 0.5)	−0.85 (−1.5, 0.2)	0.261 ^4^	
Normal	189 (78.4)	211 (81.5)	0.001 ^5^	Ref
Wasted	27 (11.2)	41 (15.8)		0.7 (0.4, 1.3)
Severely wasted	25 (10.4)	7 (2.7)		**4.0 (1.7, 9.5)**

^1^ Weight for age *z* score (WAZ) was calculated for children aged less than 121 months. ^2^ Height for age *z* score (HAZ) was calculated for children aged 0–18 years, missing data *n* = 25. ^3^ Weight for height *z* score (WHZ) was calculated for children aged less than 61 months. ^4^ Mann–Whitney U test. ^5^ Chi square test. ^6^ Adjusted for age and sex. The bold font indicates statistically significant findings.

**Table 4 nutrients-11-02728-t004:** Relationship between different types of impairments and malnutrition among children with disabilities.

Type of Impairment ^1^	Weight for Age ^2^		Prevalence of Underweight (%) (95% CI)	Adjusted OR ^4^	Height for Age ^3^		Prevalence of Stunting (%) (95% CI)	Adjusted OR ^4^
	Normal	Underweight			Normal	Stunted		
Physical impairment	157	363	69.8 (66.0–74.0)	2.7 (1.8–3.9)	236	662	73.7 (71.0–76.0)	3.4 (2.6–4.4)
Cerebral palsy (CP)	134	331	71.2 (67.0–75.0)		195	585	75.0 (72.0–78.0)	
Birth defect	14	19	57.6 (41.0–73.0)		7	5	41.7 (19.0-68.0)	
Genetic disease related physical impairment	7	10	58.8 (36.0–78.0)		26	41	61.2 (49.0–72.0)	
Trauma/Injury related physical impairment	0	2	100.0 (34.0–100.0)		5	27	84.4 (68.0–93.0)	
Musculoskeletal dystrophy/atrophy	2	1	33.3 (6.0–79.0)		3	4	57.1 (25.0–84.0)	
Visual	23	54	70.1 (59.0–79.0)	1.3 (0.8–2.2)	41	113	73.4 (66.0–80.0)	1.5 (1.0–2.2)
Hearing	66	135	67.2 (60.0–73.0)	1.2 (0.8–1.7)	121	261	68.3 (63.0–73.0)	1.2 (0.9–1.5)
Epilepsy	26	82	75.9 (67.0–83.0)	**1.9 (1.2–3.0)**	36	139	79.4 (73.0–85.0)	**2.2 (1.5–3.2)**
Speech	176	264	60.0 (55.0–64.0)	0.6 (0.4–0.8)	311	527	62.9 (60.0–66.0)	0.7 (0.5–0.9)

^1^ Type of impairments are not mutually exclusive. ^2^ Weight for age *z* score (WAZ) was calculated for children aged less than 121 months. ^3^ Height for age *z* score (HAZ) was calculated for children aged 0–18 years; missing data *n* = 25. ^4^ Adjusted for age and sex. The bold font indicates statistically significant findings.

**Table 5 nutrients-11-02728-t005:** Nutritional status of children with cerebral palsy (CP) (*n* = 796).

Clinical Characteristics	Nutritional Status of Children					
	Weight for Age ^1^		*p* Value ^3^	Height for Age ^2^		*p* Value ^3^
	Normal	Underweight		Normal	Stunted	
**Predominant Type of CP**						
Monoplegia	61 (47.7)	67 (52.3)	**<0.001**	115 (49.4)	118 (50.6)	**<0.001**
Diplegia	29 (33.0)	59 (67.0)		25 (18.5)	110 (81.5)	
Tri/Quadriplegia	26 (15.5)	142 (84.5)		34 (11.2)	270 (88.8)	
Dyskinesia	8 (25.0)	24 (75.0)		12 (24.5)	37 (75.5)	
Hypotonia	10 (23.3)	33 (76.7)		7 (14.0)	43 (86.0)	
Ataxia	0 (0.0)	0 (0.0)		0 (0.0)	1 (100.0)	
**Gross Motor Function Classification System (GMFCS) Level**						
I–II	60 (45.8)	71 (54.2)	**<0.001**	131 (52.8)	117 (47.2)	**<0.001**
III–V	72 (22.0)	256 (78.0)		61 (11.6)	464 (88.4)	
**Age of CP Diagnosis (Years)**						
Less than 2	24 (21.6)	87 (78.4)	0.093	23 (18.1)	104 (81.9)	**0.028**
2–3	47 (30.5)	107 (69.5)		47 (23.3)	155 (76.7)	
4–5	25 (29.1)	61 (70.9)		26 (20.8)	99 (79.2)	
6 and above	30 (38.5)	48 (61.5)		82 (30.5)	187 (69.5)	
**Associated Impairments**						
Visual	15 (25.9)	43 (74.1)	0.595	18 (18.6)	79 (81.4)	0.117
Hearing	30 (22.1)	106 (77.9)	**0.039**	36 (15.9)	191 (84.1)	**<0.001**
Epilepsy	23 (23.0)	77 (77.0)	0.147	26 (16.7)	130 (83.3)	**0.007**
Speech	105 (34.4)	300 (65.6)	**<0.001**	166 (30.9)	372 (69.1)	**<0.001**

^1^ Weight for age *z* score (WAZ) was calculated for children aged less than 121 months. ^2^ Height for age *z* score (HAZ) was calculated for children aged 0–18 years; missing data *n* = 25. ^3^ Chi-square test. The bold font indicates statistically significant findings.

**Table 6 nutrients-11-02728-t006:** Relationship between socioeconomic characteristics and underweight.

Characteristics	Underweight among Children with Disability *n* (%) ^1, 6^	Adjusted Odds Ratio ^2^ (95% CI)	Underweight among Children without Disability *n* (%) ^1, 7^	Adjusted Odds Ratio ^2^ (95% CI)
	*n* = 437		*n* = 234	
**Source of Drinking Water**				
Improved source ^3^	437 (64.6)	Ref	233 (100.0)	Ref
Unimproved source ^3^	0 (0.0)	N/A	0 (0.0)	N/A
**Water Treatment Method**				
Proper treatment ^4^	50 (59.5)	Ref	21 (25.6)	Ref
Improper treatment	387 (65.4)	1.3 (0.8–2.0)	212 (36.4)	1.6 (1.0–2.7)
**Access to Sanitation**				
Improved facility ^5^	314 (61.9)	Ref	166 (32.6)	Ref
Not-improved facility ^5^	123 (72.8)	**1.6 (1.1–2.4)**	67 (43.2)	**1.5 (1.0–2.2)**
**Educational Level of Mother (Total Years of Schooling)**				
0	147 (68.4)	**2.0 (1.1–3.7)**	82 (38.3)	2.5 (1.0–6.3)
1–4	71 (73.2)	**2.5 (1.3–5.1)**	45 (40.5)	**2.8 (1.1–7.4)**
5	129 (62.0)	1.5 (0.8–2.7)	71 (34.3)	2.2 (0.8–5.6)
6–9	61 (61.0)	1.5 (0.7–2.8)	30 (29.7)	1.8 (0.7–4.8)
≥10	29 (51.8)	Ref	6 (18.2)	Ref
**Educational Level of Father (Total Years of Schooling)**				
0	193 (66.1)	**1.7 (1.0–2.7)**	99 (36.4)	**2.2 (1.1–4.1)**
1–4	45 (66.2)	1.7 (0.9–3.2)	39 (41.5)	**2.9 (1.4–5.9)**
5	89 (65.9)	**1.7 (1.0–2.9)**	53 (35.8)	**2.2 (1.1–4.4)**
6–9	54 (69.2)	**2.0 (1.0–3.7)**	29 (36.7)	**2.3 (1.1–5.0)**
≥10	53 (53.5)	Ref	14 (19.4)	Ref
**Occupation of Mother**				
Desk job	3 (50.0)	Ref	0 (0.0)	0 (0.0)
Blue-collar job	23 (63.9)	1.7 (0.3–10.3)	9 (30.0)	1.4 (0.4–5.2)
Business	3 (100.0)	N/A	1 (25.0)	1.0 (0.1–12.1)
Agricultural/farming	2 (100.0)	N/A	1 (50.0)	3.5 (0.2–69.1)
Homemaker	405 (64.7)	1.8 (0.3–9.4)	218 (35.9)	1.9 (0.7–5.4)
Others	1 (33.3)	0.5 (0.03–8.9)	5 (25.0)	Ref
**Occupation of Father**				
Desk job	24 (51.1)	Ref	7 (18.9)	Ref
Blue-collar job	191 (70.5)	**2.3 (1.2–4.3)**	93 (34.1)	2.0 (0.8–4.8)
Business	78 (60.5)	1.5 (0.7–2.9)	43 (37.7)	2.4 (1.0–6.0)
Agricultural/farming	86 (63.7)	1.7 (0.8–3.3)	52 (41.6)	**2.7 (1.1–6.7)**
Unemployed	2 (50.0)	0.9 (0.1–7.3)	0 (0.0)	0 (0.0)
Others	56 (62.2)	1.6 (0.8–3.2)	39 (33.3)	2.0 (0.8–4.9)
**Socioeconomic Status (SES)**				
Poor SES	108 (70.6)	**2.0 (1.3–3.1)**	59 (41.8)	**1.7 (1.1–2.6)**
Moderate SES	206 (69.1)	**1.9 (1.3–2.7)**	99 (37.1)	1.4 (1.0–2.1)
High SES	123 (54.7)	Ref	76 (29.2)	Ref
**Monthly Income-Expenditure Balance**				
Negative balance	22 (62.9)	1.1 (0.5–2.3)	12 (40.0)	1.5 (0.7–3.4)
Equal balance	316 (66.2)	1.2 (0.8–1.8)	161 (37.1)	1.4 (1.0–2.0)
Positive balance	99 (61.5)	Ref	57 (29.7)	Ref
**Mainstream School Attendance**				
No	173 (76.5)	**3.8 (2.4–6.1)**	12 (60.0)	**2.8 (1.1–7.2)**
Yes	60 (45.5)	Ref	110 (34.6)	Ref

^1^ Weight for age *z* score (WAZ) was calculated for children aged less than 121 months. ^2^ Adjusted for age and sex. ^3^ Source of drinking water has been categorized following the definition of BDHS 2014. Improved source includes piped into dwelling, piped into year/plot, public tap, tubewell or borehole, protected well, rainwater, and bottled water; and unimproved source includes unprotected well, unprotected spring, tanker truck, surface water, and other sources. ^4^ Appropriate water treatment methods include boiling, filtering, bleaching and solar disinfecting. ^5^ Type of sanitation facility has been categorized following the definition of BDHS 2014. Improved sanitation includes flush/pour flush facilities with proper disposal system, pit latrine with slab, etc., whereas, open pit latrine, hanging toilet and no toilet facility were categorized as not-improved sanitation facility. ^6^ Row percentages of underweight were calculated for individual characteristics, % = (total number of underweight children with disability for a specific characteristic) ÷ (total number of children with disability for the same characteristic) × 100. ^7^ Row percentages of underweight were calculated for individual characteristics, % = (total number of underweight children without disability for a specific characteristic) ÷ (total number of children without disability for the same characteristic) × 100. The bold font indicates statistically significant findings.

**Table 7 nutrients-11-02728-t007:** Relationship between socioeconomic characteristics and stunting.

Characteristics	Stunting among Children with Disability, *n* (%) ^1, 6^	Adjusted Odds Ratio^2^ (95% CI)	Stunting among Children without Disability, n (%) ^1, 7^	Adjusted Odds Ratio ^2^ (95% CI)
	*n* = 820		*n* = 316	
**Source of Drinking Water**				
Improved source ^3^	820 (65.8)	Ref	315 (100.0)	Ref
Unimproved source ^3^	0 (0.0)	N/A	0 (0.0)	N/A
**Water Treatment Method**				
Proper treatment ^4^	101 (62.7)	Ref	31 (20.5)	Ref
Improper treatment ^4^	719 (66.1)	1.1 (0.8–1.6)	284 (25.5)	1.4 (0.9–2.2)
**Access to Sanitation**				
Improved facility ^5^	571 (63.7)	Ref	241 (24.1)	Ref
Not-improved facility ^5^	249 (70.7)	**1.4 (1.1–1.9)**	73 (27.9)	1.3 (0.9–1.7)
**Educational Level of Mother (Total Years of Schooling)**				
0	335 (65.4)	**2.2 (1.4–3.5)**	120 (24.4)	1.8 (0.9–3.4)
1–4	128 (71.1)	**2.7 (1.6–4.6)**	47 (23.4)	1.5 (0.7–3.0)
5	214 (65.6)	**2.0 (1.6–4.6)**	98 (26.6)	1.8 (0.9–3.4)
6–9	99 (70.2)	**2.5 (1.4–4.4)**	38 (28.4)	1.7 (0.8–3.5)
≥10	44 (48.9)	Ref	13 (18.1)	Ref
**Educational Level of Father (Total Years of Schooling)**				
0	396 (66.6)	**1.5 (1.0–2.1)**	146 (25.3)	1.6 (1.0–2.6)
1–4	90 (66.7)	1.4 (0.9–2.3)	37 (24.0)	1.3 (0.7–2.2)
5	150 (66.1)	1.4 (0.9–2.1)	65 (24.0)	1.4 (0.8–2.3)
6–9	88 (68.8)	1.4 (0.9–2.4)	39 (31.5)	1.9 (1.0–3.4)
≥10	93 (58.1)	Ref	28 (20.0)	Ref
**Occupation of Mother**				
Desk job	8 (57.1)	Ref	1 (9.1)	Ref
Blue-collar job	48 (65.8)	1.4 (0.4–4.7)	15 (30.6)	5.1 (0.6–44.2)
Business	8 (66.7)	1.7 (0.3–8.8)	0 (0.0)	N/A
Agricultural/farming	4 (66.7)	1.6 (0.2–12.0)	1 (20.0)	3.0 (0.1–62.4)
Homemaker	751 (65.8)	1.4 (0.5–4.1)	276 (30.2)	4.7 (0.6–37.6)
Others	1 (33.3)	0.3 (0.02–4.4)	23 (8.2)	1.2 (0.1–9.9)
**Occupation of Father**				
Desk job	42 (61.8)	Ref	14 (18.2)	Ref
Blue-collar job	327 (70.9)	1.6 (0.9–2.7)	123 (28.7)	**2.1 (1.1–3.9)**
Business	148 (64.3)	1.2 (0.7–2.1)	53 (25.4)	1.8 (0.9–3.6)
Agricultural/farming	188 (62.3)	1.2 (0.7–2.0)	69 (19.2)	1.4 (0.7–2.7)
Unemployed	10 (58.8)	1.0 (0.3–3.0)	1 (25.0)	1.3 (0.1–13.9)
Others	105 (61.4)	1.1 (0.6–2.0)	56 (29.3)	**2.1 (1.1–4.1)**
**Socioeconomic Status (SES)**				
Poor SES	200 (73.8)	**2.1 (1.5–2.9)**	69 (31.1)	**1.9 (1.3–2.7)**
Moderate SES	339 (67.5)	**1.5 (1.1–1.9)**	131 (28.2)	**1.5 (1.1–2.1)**
High SES	281 (59.0)	Ref	116 (19.9)	Ref
**Monthly Income-Expenditure Balance**				
Negative balance	45 (63.4)	1.0 (0.6–1.8)	10 (22.7)	1.0 (0.5–2.2)
Equal balance	610 (67.3)	**1.3 (1.0–1.7)**	234 (25.7)	1.3 (0.9–1.7)
Positive balance	162 (60.9)	Ref	69 (23.0)	Ref
**Mainstream School Attendance**				
No	430 (75.0)	**4.1 (3.0–5.4)**	12 (36.4)	2.1 (1.0–4.4)
Yes	146 (41.0)	Ref	166 (18.3)	Ref

^1^ Height-for-age *z* score (HAZ) was calculated for children aged 0–18 years, missing data *n* = 25. ^2^ Adjusted for age and sex. ^3^ Source of drinking water has been categorized following definition of BDHS 2014. Improved source includes piped into dwelling, piped into year/plot, public tap, tubewell or borehole, protected well, rainwater, and bottled water; and unimproved source includes unprotected well, unprotected spring, tanker truck, surface water, and other sources. ^4^ Appropriate water treatment methods include boiling, filtering, bleaching and solar disinfecting. ^5^ Type of sanitation facility has been categorized following the definition of BDHS 2014. Improved sanitation includes flush/pour flush facilities with proper disposal system, pit latrine with slab, etc., whereas, open pit latrine, hanging toilet and no toilet facility were categorized as not-improved sanitation facility. ^6^ Row percentages of stunting were calculated for individual characteristics, % = (total number of stunted children with disability for a specific characteristic) ÷ (total number of children with disability for the same characteristic) × 100. ^7^ Row percentages of stunting were calculated for individual characteristics, % = (total number of stunted children without disability for a specific characteristic) ÷ (total number of children without disability for the same characteristic) × 100. The bold font indicates statistically significant findings.

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
