# Peer review of "Childhood Disability and Nutrition: Findings from a Population-Based Case Control Study in Rural Bangladesh"

_nutrients, 2019, doi:10.3390/nu11112728_

Round 1

Reviewer 1 Report

This is well conducted population based study that presents interesting data about the anthropometric data for children in Bangladesh with and with disabilities.

Nevertheless, it presents data that has been previously been reported worldwide in scenarios with similar resources (i.e. Hume-Nixon, Trop Med Int Health, 2018) briefly mentioned in the discussion

It would have been more interesting to report other additional data that would help understand better the association for their malnutrition as:

Other anthropometric information (i.e. skin folds) Data on regards caloric intakeData about other comorbidities (i.e. dysphagia) because many children with disabilities have many other conditions affecting their intake and also increasing their caloric requirements.

These clear limitations should also be mentioned in detail. It was sort of commented on the 4th limitation.

Author Response

We would like to thank you for your constructive suggestions and helpful comments on our manuscript entitled “Childhood Disability and Nutrition: Findings from a Population Based Case-Control Study in Rural Bangladesh”

Please see below our point-by-point response to the comments;

Point 1: This is well conducted population-based study that presents interesting data about the anthropometric data for children in Bangladesh with and without disabilities.

Response 1: Thank you.

Point 2: Nevertheless, it presents data that has been previously been reported worldwide in scenarios with similar resources (i.e. Hume-Nixon, Trop Med Int Health, 2018) briefly mentioned in the discussion.

Response 2: Thank you for your comment. Yes, we agree that the evidence presented in this study are similar to other studies conducted in different low- and middle-income countries (LMICs).

However, as stated by Hume-Nixon et al. in their recent systematic review, such evidence is scarce in LMICs and the available studies are generally of poor quality. According to the review, inconsistency in study characteristics inhibited the investigators in reporting effect of different factors on association between childhood disability and malnutrition in LMICs. Furthermore, among the limited available studies from LMICs lack of representation from all major types of disabilities also hindered the investigation of the role of different types of disability on malnutrition among children living in LMICs.

In our study we estimated the effect of socio-demographic factors on association between disability and malnutrition among children from a large-scale, population-based, matched case-control study in rural Bangladesh. We also reported age-sex adjusted prevalence of malnutrition among children with all major types of disabilities in rural Bangladesh. Therefore, the findings are generalizable to a larger group of children. Such evidence is crucial for program planners and policy makers to design and implement disability inclusive intervention programs in Bangladesh and other LMICs.

We have edited the discussion section considering the above-mentioned points in the revised manuscript. (Line 275-295)

Point 3: It would have been more interesting to report other additional data that would help understand better the association for their malnutrition as:

Other anthropometric information (i.e. skin folds) Data on regards caloric intake, Data about other comorbidities (i.e. dysphagia) because many children with disabilities have many other conditions affecting their intake and also increasing their caloric requirements.

These clear limitations should also be mentioned in detail. It was sort of commented on the 4th limitation.

Response 3: Thank you for your valuable comment. We agree on this very important limitation of our study and we have now edited the limitation section accordingly (line 367-370).

Yours sincerely,

A/Prof Gulam Khandaker

Reviewer 2 Report

This manuscript reported the relationships between childhood disability and malnutrition in a population based database in rural Bangladesh. The results suggested that children with disability, the majorly with the form of physical impairment, had higher odds of being severely underweight (6.6), stunted (11.8) or wasted (4.0) compared to Controls. More importantly, the odds of being underweight or stunting are found to be associated with parental educational level, socio-economic status and attendance to mainstream school among children with and without disability. Overall, the manuscript is well-written and delivers the clear messages. The results are particularly important to understand the complex relationship of childhood disability, malnutrition and other contributing factors. However, the results do not imply a causal relationship between nutrition and early childhood disability.

Few minor points remained to be addressed:

Selection of controls. Age and sex matched Controls were identified by KIs (local trained volunteers). Were the KIs blinded to the individual characteristics during the selection? The authors should address the effort to avoid the selecting bias. The number of each category doesn’t appear to be correct in Figure 2. For example, the number of children categorized as underweight in children with disability is seven. This number seems to be a lot lower than that in Table 3 despite the results are reported in children with different age range. The stats for the general, acute, and chronic malnutrition are determined in three different age ranges without considering the whole cohort (Table 3). Please address the rational.

Author Response

We would like to thank you for your constructive suggestions and helpful comments on our manuscript entitled “Childhood Disability and Nutrition: Findings from a Population Based Case-Control Study in Rural Bangladesh”

Please see below our point-by-point response to the comments;

Point 1: This manuscript reported the relationships between childhood disability and malnutrition in a population-based database in rural Bangladesh. The results suggested that children with disability, the majorly with the form of physical impairment, had higher odds of being severely underweight (6.6), stunted (11.8) or wasted (4.0) compared to Controls. More importantly, the odds of being underweight or stunting are found to be associated with parental educational level, socio-economic status and attendance to mainstream school among children with and without disability. Overall, the manuscript is well-written and delivers the clear messages. The results are particularly important to understand the complex relationship of childhood disability, malnutrition and other contributing factors.

Response 1: Thank you.

Point 2: However, the results do not imply a causal relationship between nutrition and early childhood disability.

Response 2: Thank you for your thoughtful comment. It is true that we did not explore the causal relationship between early childhood disability and malnutrition. We aimed to identify the underlying factors that increases vulnerability of children with disability living in rural Bangladesh.

However, this is an important inherent limitation of our study (as an observational study) and we have now edited the limitation section accordingly (line 367-370).

Point 3: Selection of controls. Age and sex matched Controls were identified by KIs (local trained volunteers). Were the KIs blinded to the individual characteristics during the selection? The authors should address the effort to avoid the selecting bias.

Response 3: Thank you for your valuable comment on this very important issue. The KIs were not aware about any of the individual characteristics (except for age and sex) during the selection of controls. During the key informant (KI) training, they were trained to select two controls randomly for each identified case who meet following criteria,

Have no forms of disability Same age and sex of the child with disability Lives in the neighborhood (within 10-20 households) of the child with disability

Following KI training, all 130 KIs randomly selected the controls from study area based on above mentioned selection criteria. Hence it is unlikely to impose any selection bias during control selection.  

We have rephrased the statement for clarity (line 104-108)

Point 4: The number of each category doesn’t appear to be correct in Figure 2. For example, the number of children categorized as underweight in children with disability is seven. This number seems to be a lot lower than that in Table 3 despite the results are reported in children with different age range.

Response 4: Thank you for your keen observation and the correction. It was an unintentional mistake. Our sincere apologies. The figure 2 illustrates nutritional status of children aged 0-61 months with or without disability. We have now corrected the numbers. In figure 2.

Point 5: The stats for the general, acute, and chronic malnutrition are determined in three different age ranges without considering the whole cohort (Table 3). Please address the rational.

Response 5:  Thank you for your thoughtful comment. General undernutrition was defined using weight for age z score which was calculated only for children aged 0-121 months as WHO Anthroplus does not calculate z scores above that age group. The justification for that is after 121 months, the weight of a child is also highly affected by their height. The chronic undernutrition as defined by height for age z score was calculated for all aged groups using WHO AnthroPlus software, however, there were few missing values (height and/or age), therefore the total number did not add-up to the total study population. For acute undernutrition we used weight for height z scores which was calculated using WHO Anthro software. However, WHO Anthro software calculates weight for height z scores only for children aged upto 61 months. Hence, we could not present the acute malnutrition in older groups.

We have edited the footnote in table 3 and also elaborated these in our limitation section (line 363-367).

Yours sincerely,

A/Prof Gulam Khandaker